# Latest News from the “Guardian”: p53 Directly Activates Asymmetric Stem Cell Division Regulators

**DOI:** 10.3390/ijms26073171

**Published:** 2025-03-29

**Authors:** Ana Carmena

**Affiliations:** Instituto de Neurociencias, Consejo Superior de Investigaciones Científicas/Universidad Miguel Hernández de Elche, Sant Joan d’Alacant, 03550 Alicant, Spain; acarmena@umh.es

**Keywords:** p53, asymmetric stem cell division, neural stem cells, *Drosophila*, Numb/NUMB, Brat/TRIM proteins, Traf4/TRAF4

## Abstract

Since its discovery in 1979, the human tumor suppressor gene *TP53*—also known as the “guardian of the genome”—has been the subject of intense research. Mutated in most human cancers, TP53 has traditionally been considered a key fighter against stress factors by trans-activating a network of target genes that promote cell cycle arrest, DNA repair, or apoptosis. Intriguingly, over the past years, novel non-canonical functions of p53 in unstressed cells have also emerged, including the mode of stem cell division regulation. However, the mechanisms by which p53 modulates these novel functions remain incompletely understood. In a recent work, we found that *Drosophila* p53 controls asymmetric stem cell division (ASCD) in neural stem cells by transcriptionally activating core ASCD regulators, such as the conserved cell-fate determinants Numb and Brat (NUMB and TRIM3/TRIM2/TRIM32 in humans, respectively). In this short communication, we comment on this new finding, the mild phenotypes associated with *Drosophila p53* mutants in this context, as well as novel avenues for future research.

## 1. The “Guardian of the Genome” and More: Newly Emerged p53 Functions

More than 30 years ago, one of the discoverers of p53 [1,2,3], Professor Sir David Lane, commented that “research on the p53 protein is at fever pitch” [4]; after all these years, it seems that we have not been able to bring this fever down. Today, we know that the tumor suppressor *TP53* is mutated in more than 50% of human tumors [5,6]. Many *TP53* mutations not only lose the tumor suppressor properties of p53 but also display gain-of-function oncogenic features [5,7]. However, under normal conditions, TP53 is a crucial regulator of multiple cellular stress conditions, such as acute DNA damage, hypoxia, and oncogene signaling, among others [8]. In the absence of cellular stress, TP53 levels are very low, as it is targeted for degradation by the E3 ubiquitin ligase MDM2 [9]. The presence of cellular stress factors promotes the stabilization of TP53, which in turn activates target genes that have traditionally been shown to induce apoptosis, senescence, cell-cycle arrest, or DNA repair [10]. Remarkably, over the past decades, different studies have unveiled novel pathways, cellular processes, and target genes regulated by p53 in unstressed cells [8,11,12,13,14,15,16,17,18,19,20,21,22,23]. Among these newly emerged non-canonical functions of p53, one that is particularly intriguing is the mode of stem cell division regulation [11,18,22]. In the original study describing this p53 novel function, the authors showed how, in mammary stem cells, p53 restrains symmetric self-renewing divisions in favor of asymmetric, differentiation-driven ones. However, the mechanism by which p53 regulates this process remains unclear [11].

## 2. Asymmetric Stem Cell Division

In an asymmetric stem cell division (ASCD), two different daughter cells are generated; one daughter cell keeps on self-renewing, like the mother stem cell, while the other initiates a differentiation program. Hence, ASCD is a crucial process in generating cell diversity during development, as well as in regulating adult tissue homeostasis [24]. For many decades, the neural stem cells of *Drosophila*, called neuroblasts (NBs), have been used as an excellent paradigm for gaining deeper insight into the ASCD process [25]. NBs divide asymmetrically, giving rise to another NB (the stem-like cell) and a daughter cell that stops self-renewing. In the case of type I NBs (NBI), this latter cell is called a ganglion mother cell (GMC), which will divide just once asymmetrically to give rise to two neurons or glial cells. In the case of type II NBs (NBII), this daughter cell is called an intermediate neural progenitor (INP), which will divide asymmetrically to give rise to another INP (that will continue dividing asymmetrically for a limited number of divisions) and a GMC [25] (Figure 1a). The generation of two different daughter cells implies the coordinated action of an intricate network of regulatory proteins. Specifically, during NB division, at metaphase, a protein complex formed by multiple modulators, including small GTPases, Par proteins, and aPKC, is located at the apical pole of the NB [24,26,27,28,29,30,31,32,33,34,35,36,37,38,39,40,41]. These apical proteins (also known as “the apical complex”) are responsible for two key processes in an asymmetric NB division: (1) exclusion to the basal NB cortex of the so-called cell-fate determinants and (2) the correct orientation of the mitotic spindle along a previously established apical–basal axis of cell polarity (Figure 1b). In this way, when the NB divides, the cell-fate determinants will segregate exclusively to the most basal daughter cell, repressing in this cell the self-renewal program. Only a few cell-fate determinants have been described in *Drosophila* NBs; among these are the conserved proteins Numb and Brain Tumor (Brat) [25,42,43,44,45,46,47,48,49]. 

### 2.1. The Cell-Fate Determinants Numb and Brat

Originally identified in *Drosophila* as the first determinant required to confer different cell fates to the progeny of asymmetrically dividing sensory organ precursors [47,48], the membrane-associated protein Numb was promptly proven to be highly conserved in mammals [50,51,52,53,54,55,56,57]. Likewise, the initial characterization of Numb as a key repressor of the transmembrane receptor Notch in *Drosophila* was also shown to be a conserved feature in mammals, with relevant consequences in human tumors [51,58,59,60,61,62]. Since then, however, Numb has been revealed as a versatile and active (“not-Numb”) regulator involved in modulating numerous pathways and cellular processes, including endocytosis and ubiquitination [63,64]. In fact, one of the newly characterized functions of human NUMB is as a positive regulator of TP53, as Numb avoids MDM2-mediated ubiquitination and consequent TP53 degradation [65]. In mouse epithelial cells, Numb also promotes high p53 activity in the cell in which it is asymmetrically segregated, and Numb loss causes p53-dependent tumorigenesis, as restored p53 levels and activity decrease over-proliferation [66]. Therefore, the Numb–p53 circuitry seems to be very relevant in these contexts [67].

The function of the *Drosophila* cell-fate determinant Brat in ASCD was identified years later than that of Numb [42,43,44,45]. Brat is related to the human tripartite-motif (TRIM) proteins TRIM3, TRIM2, and TRIM32; however, it is an atypical TRIM-NHL protein, as Brat lacks the RING domain characteristic of this family [68,69]. As in the case of mammalian Numb, a regulatory connection between TRIM proteins and p53 has been established [70,71]. For example, most TRIM proteins have E3-ligase activity that can affect the stability of p53, promoting its degradation through the ubiquitin–proteasome system. This is the case of TRIM-32, which, in turn, is a target of p53 under stress conditions; in this way, a negative feedback loop that diminishes p53 function is established [72]. Thus, both Numb–p53 and TRIM proteins–p53 regulatory axes are worthy of further exploration. In *Drosophila*, during NB ASCD, Brat—along with Numb and the other main cell-fate determinant described in *Drosophila*, the transcription factor Prospero (PROX1 in humans)—represses self-renewal and induces differentiation in the daughter cell in which it is segregated, and all of them can induce tumor-like overgrowth when compromised in this context [44,73] (see also below).

### 2.2. ASCD Failures Can Cause Tumor-like Overgrowth

In 2005, it was shown for the first time that failures in the ASCD process can lead to tumorigenesis using the *Drosophila* larval central brain NBs as an experimental model [74]. At around the same time, *Drosophila* genes originally identified as tumor suppressor genes [75,76,77], such as *discs large 1* (*dlg1*), *lethal (2) giant larvae* (*l(2)gl*), and *brain tumor* (*brat*), were shown to be involved in ASCD regulation, further supporting the connection between ASCD malfunction and tumorigenesis [42,43,44,45,78,79,80,81]. Remarkably, failures in some NB ASCD regulators do not cause tumor-like overgrowth [82,83]. For example, larval brain NB mutant clones for the ASCD regulators *canoe* (*cno)*, *scribble (scrib)*, *dlg1*, or *l(2)gl* display ectopic NBs, but they do not overgrow. However, the simultaneous loss of *scrib* and *cno* in NB clones leads to the formation of tumoral masses [82]. In fact, it appears that only mutations in these determinants—the “last effectors” of the ASCD process—induce tumor-like overgrowth, while mutations in most, if not all, the ASCD apical regulators are compensated by other functional regulators, thus avoiding tumor formation [82,83,84]. Despite this, the presence of these single ASCD regulator mutations confers a higher susceptibility to tumor development in cases where additional mutational events in genes encoding ASCD modulators occur. Altogether, the knowledge accumulated over the past decades strongly indicates that ASCD regulators behave as tumor suppressor genes. Hence, one possibility was that well-known tumor suppressor genes could be acting as ASCD modulators. This hypothesis led us to investigate the potential role in ASCD of one of the most relevant tumor suppressor genes: p53. 

## 3. *Drosophila* p53 Directly Activates Key ASCD Regulators: Why So Mild Phenotypes?

In a recent report, we observed neuron losses or duplications in NBI lineages of *p53* null mutant embryos, a phenotype reminiscent of that observed in ASCD regulator mutations. The percentage of failures found, although significant, was not very high; this is also a characteristic feature detected in mutants for ASCD regulators [40,41,85]. Moreover, the localization of key ASCD regulators, such as the cell-fate determinants Numb and Brat, was altered in embryonic metaphase NBs of *p53* mutants. The phenotype was robust but not completely penetrant: only about 30% of the NBs analyzed displayed these localization defects. However, again, this is a peculiarity of the phenotype observed in other ASCD regulator mutants [30,40,82,86]. One intriguing aspect of this latter phenotype, especially in the case of Numb mislocalization, was that, in most cases (96,1% of the metaphase NBs analyzed), the defect was the “absence” of Numb. Given that p53 is a transcription factor, one straightforward possibility was that p53 was directly activating these ASCD regulators. Interestingly, a meta-analysis of transcriptomic and ChIP-seq datasets unveiled a conserved set of human and mice predicted p53 target genes [87], and the counterparts in *Drosophila* of some of these target genes are known ASCD regulators. This is the case for the human genes *TRIM32* and *TRAF4,* which are related to *Drosophila brat* and the apical ASCD regulator *Traf4*, respectively [42,43,45,88]. Thus, we decided to focus on *Drosophila numb, brat*, and *Traf4* to determine whether they were targets of *Drosophila* p53. Chip-seq experiments revealed that *Drosophila* p53 binds and activates the expression levels of all three genes (Figure 2) [85]. 

This was an exciting discovery: we had found a mechanism by which p53, at least in *Drosophila*, regulates the mode of stem cell division. Remarkably, regardless of the low sequence conservation and evolutionary distance, *Drosophila p53* is considered the functional homolog of human *TP53* [89,90]. For example, both are important inductors of apoptosis [91,92]. Furthermore, even though *Drosophila* p53 is not involved in DNA-damage-induced cell-cycle arrest, like human TP53, it does regulate cell-cycle progression in particular metabolic stress scenarios, such as mitochondria dysfunction [91,92,93]. In addition, despite the absence of a homolog of the human E3 ubiquitin ligase MDM2 in *Drosophila*, other functionally equivalent ubiquitin ligases or negative regulators of *Drosophila* p53 have been described [91,94,95]. Hence, it would be appealing to investigate whether mammalian/human p53 also directly activates *NUMB*, the closest mammalian target (out of the three: *NUMB*, *TRAF4*, and *TRIM32*) associated with the mode of stem cell division regulation [11,52,54,57,58]. *TRIM3,* which belongs to the same family as *TRIM32*, would also be interesting to consider as a potential target of mammalian/human p53, as this tumor suppressor has been shown to regulate ASCD in humans [96].

Returning to the *Drosophila p53* phenotype, given that we found that p53 is a novel ASCD regulator, we wondered whether p53 loss could cause tumor-like overgrowth in larval brain NB lineages. However, despite the fact that p53 activates key cell-fate determinants—such as Numb and Brat, whose loss induces tumorigenesis in these lineages—*p53* mutant NB lineages do not show any overgrowth [44,73,85]. The most plausible explanation for this—and for all the other previously mentioned mild *p53* mutant phenotypes—is the significant redundancy in the ASCD process regulation to basally localize the cell-fate determinants. The asymmetric localization of these determinants in the mother stem cell, and their exclusive segregation to one of the daughter cells, is crucial in arresting self-renewal in this daughter cell. Thus, a complex network of proteins and checkpoint mechanisms has evolved to robustly ensure the successful regulation of this process. Our current hypothesis is that p53 functionally interacts with other ASCD regulators to activate the cell-fate determinants Numb and Brat. Thus, in *p53* mutants, these regulators might even be “over-active” in order to compensate for the decreased Numb and Brat levels; in this way, a strong, tumor-like overgrowth phenotype is avoided. In fact, as explained above, in embryonic *p53* mutant dividing NBs, Numb and Brat were still present and correctly located in about 70% of the NBs analyzed. This result supports the idea that additional mechanisms are activated in *p53* mutants to compensate for this loss. This phenomenon has been previously reported [41,82,83,86]. For example, only the simultaneous loss of two ASCD regulators, Cno and Scrib, leads to a complete penetrant phenotype (100%) regarding Numb or aPKC (an apical ASCD regulator) mislocalization and to tumor-like overgrowth. Each of the single mutants only shows partially penetrant phenotypes concerning Numb and aPKC distribution and do not show any tumoral overgrowth [82]. Hence, the search for potential mechanisms/regulators that interact with p53 will be an important challenge for future studies aiming to untangle the biological significance of p53 function in the context of ASCD.

## 4. Future Directions 

*Drosophila* p53 can help to enlighten some aspects of human TP53 function. In fact, *Drosophila* p53 has importantly contributed to unveil novel non-canonical p53 functions, such as tissue growth coordination, metabolic homeostasis, and cell competition [20,97,98]. Thus, it would be engaging to determine whether TP53 also regulates the mode of stem cell division by directly activating the mammalian homologues of *numb*, *brat*, and *Traf4*, the *Drosophila* p53 target genes found in this context. It would be especially intriguing to analyze whether TP53/Trp53 transcriptionally activates *TRIM3* and *NUMB*, which encode the proteins most closely involved in regulating the mode of stem cell division in mammals. This would add another layer to our comprehension of the Numb–p53 and TRIM protein–p53 regulatory axes. Similarly, we should take advantage of the amenability of *Drosophila* as a model system in order to screen for ASCD regulators that might synergistically interact with p53 in modulating the mode of stem cell division. This will contribute to enhancing our knowledge regarding the multilayered tumor suppressor activity of human TP53.

## Figures and Tables

**Figure 1 ijms-26-03171-f001:**
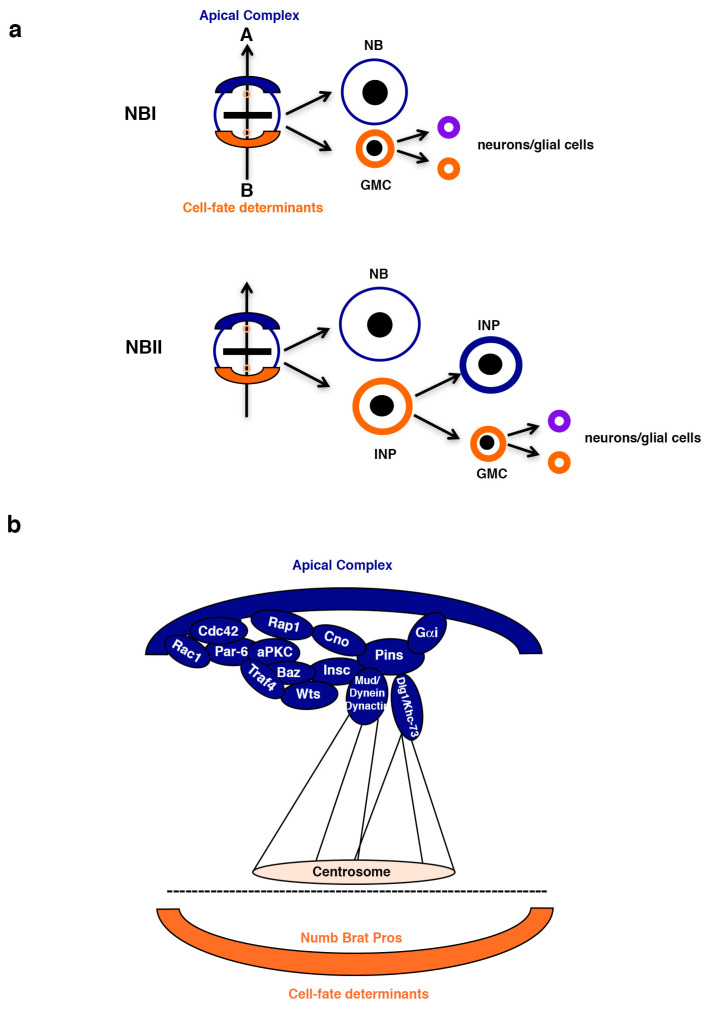
Asymmetric division of *Drosophila* NBs. (**a**) *Drosophila* NBIs divide asymmetrically to give rise to another NB and a GMC that receives the cell-fate determinants (orange) distributed at the basal pole of the mother NB. The GMC divides once more asymmetrically to give rise to two neurons or glial cells. An “apical complex” of regulators (blue) is located at the apical pole of the mother NB. This complex excludes basally the cell-fate determinants and orientates the mitotic spindle along the apical (A)–basal (B) axis. *Drosophila* NBIIs divide asymmetrically to give rise to another NB and an INP that continues dividing asymmetrically to, in turn, give rise to another INP and a GMC. NB: neuroblast; GMC: ganglion mother cell. (**b**) A simplified diagram of the intricate protein network that forms the “apical complex”, which includes small GTPases (Rac1, Cdc42, and Rap1), Par proteins (Par-6 and Bazooka, Baz/Par-3) and atypical Protein Kinase C (aPKC), among others. Cno: Canoe; Insc: Inscuteable; Pins: Partner of Inscuteable; Wts: Warts; Mud: Mushroom body defect; Dlg1: Discs large 1; Gαi: G protein alpha subunit i (This figure was made using PowerPoint and Adobe Photoshop CS6).

**Figure 2 ijms-26-03171-f002:**
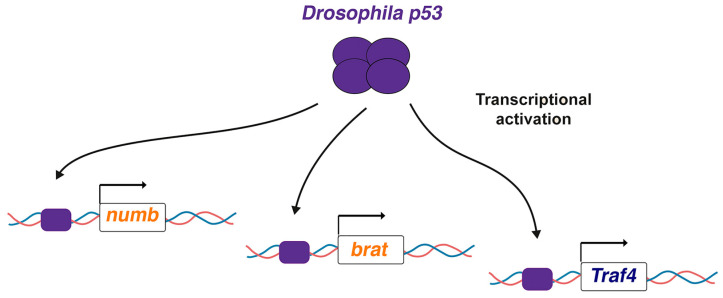
*Drosophila* p53 transcriptionally activates ASCD regulators. *Drosophila* p53 directly binds to the regulatory regions of *numb*, *brat*, and *Traf4*, which encode cell-fate determinants (Numb and Brat) and an apical complex regulator (Traf4). Adapted from [85].

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
