# Peer review of "Latest News from the “Guardian”: p53 Directly Activates Asymmetric Stem Cell Division Regulators"

_ijms, 2025, doi:10.3390/ijms26073171_

Round 1
Reviewer 1 Report
Comments and Suggestions for Authors
The manuscript "The "Guardian" has news: p53 directly activates asymmetric stem cell division regulators" sent to “International Journal of Molecular Sciences". In which the author analyzes the participation of the p53 protein in controlling the asymmetric division of stem cells by acting on proteins such as Numb and Brat. The paper is well written, the objective is clear, however I have some questions and comments to the author which I list below:
- “….using the Drosophila larval central brain NBs as an experimental system.” I considering that is experimental model, ¿why the author thinks this is system?
- Was figure 1 made by the author? If the answer is yes, you should indicate in which program or application it was made.
- “Drosophila p53 directly activates key ASCD regulators: why so mild phenotypes?” What does the author mean by a weak phenotype?
- The author comments that it is important to study the effect of p53 with Numb and TRIM to regulate tumor suppressor activity, however, the literature shows how studies have been performed with these proteins in various types of cancer. What can the author comment on this?

Author Response
Please see the attachment with the Response to the reviewer comments

Reviewer 2 Report
Comments and Suggestions for Authors
According to the editor’s strict regulation, I have carefully read and checked the article described by Carmena based on its scientific significance, soundness and novelty. In the present review article, the author overviewed the representative tumor suppressor p53-mediated ASCD. In addition to the classical tumor-suppressive function, the author described that p53 plays a pivotal role in the regulation of ASCD through direct transactivation of mammalian homologues of numb and brat. Although the present study might have certain impact on the related field, there are several concerns (see below) which should be adequately addressed before reconsideration.
Minor concerns
What is the most evident difference(s) between mammalian and Drosophila p53?
<2.2. ASCD failures can cause tumor-like overgrowth>
The author mentioned “Drosophila genes originally identified as tumor-suppressor genes”. For the convenience of the readers, the author has to describe the nature of them.
The author suggests that well-known tumor suppressor genes could be acting as ASCD modulators. This is also the case in human?
The author described that there exists a strong link between disfunction of ASCD regulators (strong determinants) and tumorigenesis in Drosophila, whereas dysregulation of most ASCD regulators (weak determinants) could be compensated by other functional modulators, which might contribute to the evasion of tumor formation. What is the difference between strong and weak determinants?
< Drosophila p53 directly activates key ASCD regulators: why so mild phenotypes?>
Drosophila p53 do bind and activates the expression levels of all three genes. The promoter regions of these three genes carry the putative p53-responsive element?
The author has to discuss how mutant p53 could affect subcellular localization of Numb.
It is unclear whether the author plans to elucidate the molecular mechanisms behind the action of p53 under Drosophila system, or apply the results obtained from Drosophila system to the better understanding of human diseases such as cancer.
For the convenience of the specialized and non-specialized readers, English writing should be further improved.
Comments on the Quality of English Language
For the convenience of the specialized and non-specialized readers, English writing should be further improved.
Author Response
Please see the attachment with the Response to the Reviewer's comments
